

# Transport into the polar stratosphere from the Asian monsoon region

Xiaolu Yan[1], Paul Konopka[2], Felix Ploeger[2,3], and Aurélien Podglajen[4]

[1]State Key Laboratory of Severe Weather & Institute of Tibetan Plateau Meteorology, Chinese Academy of Meteorological Sciences, Beijing, China
[2]Institute for Energy and Climate Research: Stratosphere (IEK-7), Forschungszentrum Jülich, Jülich, Germany
[3]Institute for Atmospheric and Environmental Research, University of Wuppertal, Wuppertal, Germany
[4]Laboratoire de Météorologie Dynamique (LMD/IPSL), École polytechnique, Institut polytechnique de Paris, Sorbonne Université, École normale supérieure, PSL Research University, CNRS, Paris, France

**Correspondence:** Xiaolu Yan (xiaoluyan@cma.gov.cn) and Paul Konopka (p.konopka@fz-juelich.de)

**Abstract.** The South-East Asian boundary layer has witnessed alarming pollution levels in recent years, which even affects the trace gas composition in the southern hemisphere by inter-hemispheric transport. We use $SF_6$ observations and the Lagrangian chemistry transport model CLaMS, driven by the ERA5 reanalysis data for the period 2010-2014, to assess the impact of the Asian monsoon (AM) region [15° N, 45° N, 30° E, 120° E] as a significant source of pollutants for the stratosphere, in particular in polar regions. We examine the contribution of transport from the AM region to the Northern Hemisphere polar region (NP) [60° N, 90° N] and to the Southern Hemisphere polar region (SP) [60° S, 90° S]. Despite the smaller geographical size of the AM region when compared to the Southern Hemisphere subtropics [15° S, 45° S] and tropics [15° S, 15° N], our findings reveal that the air mass fractions from the AM to the polar regions are approximately 1.5 times larger than the corresponding contributions from the Southern Hemisphere subtropics and roughly two times smaller than those from the tropics. The transport of air masses from the AM boundary layer to the stratospheric polar vortex primarily occurs above an altitude of about 450 K and over timescales exceeding 2 years. In contrast, transport timescales to the polar regions situated below the vortex are shorter, typically less than about 2 years. Furthermore, the transport contribution from the AM region to the polar regions exhibits distinctive inter-annual variability, significantly influencing the distributions of pollutants. Our analysis of detrended $SF_6$ from ACE-FTS over the polar regions reveals a strong correlation with the fraction of relatively young air (less than two years old) originating from the AM, Southern Hemisphere subtropics, and tropics. Importantly, our reconstructed $SF_6$ data indicates that approximately 20% of $SF_6$ in both the northern and southern polar stratosphere originates from the AM boundary layer. The largest fraction of $SF_6$ in the polar stratosphere still originates from the tropical boundary layer, contributing about 50% of $SF_6$.

## 1 Introduction

Over the past few decades, rapid economic development in South-East Asia has been associated with a notable increase in the emissions of various pollutants, including $SO_2$, $NO_x$, ozone precursors, particulate matters, and others (e.g. Granier et al.,



2011; Kurokawa et al., 2013; Anenberg et al., 2019; Tessum et al., 2022). According to the Emissions Database for Global Atmospheric Research (EDGAR), South-East Asia reports the largest emissions of the three major long-lived greenhouse gases ($CO_2$, $CH_4$, and $N_2O$) when compared to other continents (Janssens-Maenhout et al., 2019). Additionally, data from
TROPOMI and CAMS reveal elevated total CO column values and anthropogenic CO emission inventories over Asia (e.g. Wang et al., 2022).

The elevated emissions of pollutants within the South-East Asian boundary layer have potential impacts on the chemical composition of the atmosphere, thereby influencing atmospheric chemistry, radiative properties, and climate (e.g. Rosenlof et al., 1997; Berntsen et al., 1999; Li et al., 2005; Park et al., 2009; Solomon et al., 2010; Riese et al., 2012; Chirkov et al., 2016;
Santee et al., 2017; Rolf et al., 2018). These impacts extend beyond the local boundary layer, significantly affecting the upper troposphere and lower stratosphere (UTLS) (e.g. Dethof et al., 1999; Vernier et al., 2011; Vogel et al., 2016; Fadnavis et al., 2018). Particularly, the Asian summer monsoon (ASM) circulation, which dominates the UTLS in the Northern Hemisphere (NH) during boreal summer, serves as an efficient pathway for transporting pollutants from the Asian boundary layer to the stratosphere over Asia and the tropics (e.g. Randel et al., 2010; Yu et al., 2017; Bian et al., 2020). Through this transport path-
way, SE Asian emissions might affect the atmospheric composition inside the ASM anticyclone and in the tropical lower stratosphere. These changes may then propagate horizontally along isentropic surfaces and vertically through the Brewer–Dobson circulation, ultimately influencing the stratosphere, even at high latitudes (e.g. Park et al., 2013; Vogel et al., 2015; Pan et al., 2016; Ploeger et al., 2017).

Numerous observational studies have provided compelling evidence of the presence of high levels of tropospheric tracers
within the ASM anticyclone, including water vapor, CO, and aerosols (e.g. Park et al., 2009; Vernier et al., 2011; Santee et al., 2017). Several previous studies have investigated the transport of ASM origin tracers to the global stratosphere (e.g. Wright et al., 2011; Bergman et al., 2012; Orbe et al., 2015), all of which underscore the dominant role played by the ASM in transporting tropospheric air into the stratosphere. Furthermore, some other studies aimed to quantify the specific contributions of tracers originating from the ASM anticyclone to the stratosphere. For instance, Garny and Randel (2016) calculated trajectories origi-
nating from the ASM anticyclone and found that 48% of these trajectories reach the stratosphere within 60 days, with 24% and 15% of trajectories subsequently transported to the tropics and the lower stratosphere of the NH, respectively. Consequently, the ASM anticyclone contributes 14% and 29% of water vapor to the tropical pipe and to the NH extratropical lower stratosphere, respectively (Nützel et al., 2019).

Many previous studies have primarily focused on assessing the transport from the Asian monsoon (AM) region to the tropics
and the NH (e.g. Tzella and Legras, 2011; Orbe et al., 2015; Yu et al., 2017). However, relatively little scientific attention has been devoted to understanding the contribution of the AM boundary layer to the polar stratosphere and its quantitative impact on pollutants, particularly in the southern polar stratosphere. Montzka et al. (2018) reported an unexpected global increase in the levels of ozone-depleting CFC-11, primarily attributed to a rise in emissions from eastern Asia after 2012. Ploeger et al. (2017) released artificial tracers from the ASM anticyclone level and found a strong correlation between ASM origin air and
HCN observations from the ACE-FTS satellite instrument. Yan et al. (2019) found that a greater quantity of ASM origin air





is transported to the Southern Hemisphere (SH) compared to the NH after 9 months, when the simulated tracers are released below the tropopause.

Motivated by these previous studies, our research particularly aims at examining the transport of air masses from the AM region into the stratosphere. This investigation is particularly crucial due to the sensitive climate effect associated with anthro-
pogenic activity and the complex atmospheric chemical effect related to the ozone hole over polar regions (e.g. Clem et al., 2020; World Meteorological Organization, 2022). Our primary goal is to quantitatively assess the air mass fraction (AMF) originating from the AM region that reaches the polar stratosphere and its impact on pollutants. To achieve this goal, we utilize ERA5 reanalyses to drive the Chemical Lagrangian Model of the Stratosphere (CLaMS), enabling us to analyze AMFs and transit times from the source region into the polar stratosphere and to validate the simulations. Additionally, we reconstruct $SF_6$
data through a combination of surface boundary measurements and model simulations, facilitating the quantification of $SF_6$ transport contributions from the AM region to polar regions. Section 2 presents the data and methods used for our analyses. In Sect. 3, we explore the seasonality of transport from the AM region and other source regions, along with a comprehensive diagnosis of the transport into polar region. We investigate the influence of transport from the AM region on the $SF_6$ and assess the quantitative contributions in Sect. 4. Finally, we discuss our findings in Sect. 5, followed by a summary of the key results
in Sect. 6.

## 2 Data and methods

In this study, we employ the CLaMS model to calculate age spectra and AMF to investigate transport processes originating from the surface of the AM region, defined as the area spanning 15-45°N and 30-120°E. CLaMS, a Lagrangian chemistry transport model (CTM), simulates trace gas transport driven by horizontal winds and total diabatic heating rates derived from reanalysis
data (e.g. McKenna et al., 2002; Konopka et al., 2004; Pommrich et al., 2014). For comparison, we divide the Earth's surface into five distinct regions to compare transport contributions. These regions include the Northern Hemisphere extratropics (extNH: 45-90° N), Northern Hemisphere subtropics (subTR-NH: 15-45° N), tropics (15° S-15° N), Southern Hemisphere subtropics (subTR-SH: 15-45° S), and Southern Hemisphere extratropics (extSH: 45-90° S). The source domains and destination regions for this study are listed in Table 1.

**Table 1.** List of all source regions used in the model simulations and destination regions investigated by this study.

| Source | Latitude | Longitude | Destination | Latitude | Longitude |
|--------|----------|-----------|-------------|----------|-----------|
| AM | 15-45°N | 30-120°E | polarLS-NH | 60-90° N | 0-360° |
| extNH | 45-90° N | 0-360° | polarLS-SH | 60-90° S | 0-360° |
| subTR-NH | 15-45° N | 0-360° | | | |
| tropics | 15° S-15° N | 0-360° | | | |
| subTR-NH | 15-45° S | 0-360° | | | |
| extSH | 45-90° S | 0-360° | | | |



We apply the Boundary Impulse (time-) Evolving Response (BIER) approach to calculate the age spectrum following the methodology outlined in Ploeger and Birner (2016). The BIER approach is an extension of the Boundary Impulse Response (BIR) method (e.g. Haine et al., 2008; Li et al., 2012), such that the method evolves with time in a transient simulation. The approach in this work is similar to our previous study in Yan et al. (2021), albeit focusing on different source regions. We initiate multiple tracer pulses within the respective boundary source regions, denoted as $\Omega_i$, with $i$ representing the source

domain (e.g., AM, NH extratropics, NH subtropics, tropics, SH subtropics, SH extratropics). The passive tracer, characterized by its mixing ratio $\chi_i$ at location $r$ and time $t$, is related to the mixing ratio $\chi_0(t)$ originating from the boundary surface of various source regions. This relationship defines the AMF from the respective source regions and can be expressed as follows:

$$\chi_i(r,t) = \int_0^\infty \chi_0(\Omega_i, t-\tau) G(r,t|\Omega_i, t-\tau) d\tau \qquad (1)$$

The age spectrum $G$ is computed by releasing 240 pulses of inert trace gas species from six distinct source regions, with

each region pulsing 40 different species. These pulse tracers approximate a delta distribution lower boundary condition $\chi_0^j$ ($\Omega_i$, t)=$\delta(t-t_j)$, where $j$ ranges from 1 to 40, defining the tracer pulses at specific source times $t_j$. The pulse tracer mixing ratios are initially set to one within the boundary layer of the source region for a duration of 30 days. Outside of the initialization region, these mixing ratios in the boundary are set to zero in each time step. To elaborate further, the first 24 different species ($j$=1,...,24), characterized by transit times of less than 2 years, are pulsed every month. Subsequently, the remaining 16 different

species ($j$=25,...,40) are pulsed every sixth month (e.g., the 25th species is pulsed in the 30th month, the 26th species in the 36th month, and so on). Consequently, all species have been pulsed after 10 years of model simulation. Ten years after release, the pulse tracer mixing ratio is set to zero again throughout the entire atmosphere and subsequently pulsed again. As a result, the model simulations provide a monthly resolution age spectrum for transit times shorter than two years and a semi-annual resolution age spectrum for longer transit times. For additional details regarding the model setup and the calculation of age

spectra from multiple pulse tracers, please refer to Ploeger and Birner (2016), Ploeger et al. (2019), Podglajen and Ploeger (2019), and Yan et al. (2021).

To quantify the contributions of surface origin air from the six regions to stratospheric pollutants, we carry out the reconstruction of $SF_6$ data using age spectrum simulations in conjunction with surface composition observations from National Oceanic and Atmospheric Administration (NOAA) (Dutton et al., 2024). $SF_6$, characterized by an atmospheric lifetime of

approximately 3200 years, remains virtually unaltered in the atmosphere for many centuries following its emission. Consequently, $SF_6$ serves as a valuable tracer for evaluating transport in models and calculating mean ages (e.g. Denning et al., 1999; Waugh et al., 2013). The reconstructed $SF_6$, characterized by its mixing ratio at location $r$ and time $t$, is related to the $SF_6$ concentration emitted from the boundary layer of various source regions. This defines the reconstructed $SF_6$ concentration transported from the respective source regions and can be calculated following Equation 1. However, in the case of $SF_6$, $\chi_0(t)$

is the $SF_6$ mixing ratio at the boundary layer in the respective surface patch.

We employ observations of $SF_6$ obtained from the Atmospheric Chemistry Experiment Fourier Transform Spectrometer (ACE-FTS) satellite instrument (e.g. Bernath et al., 2005) to validate the reconstructed $SF_6$ data. In this study, our simulation





encompasses the period from 2000 to 2014, driven by meteorological data from ERA5 reanalyses. Due to the 10 year spin-up time required for the air mass and age spectra, we focus on analyzing the model data and ACE-FTS observations specifically

from 2010 to 2014 in the following section. It is worth noting that the truncation of calculated age spectra to 10 years could cause a high-bias in reconstructed $SF_6$ mixing ratios in the stratosphere. This timeframe aligns with the period analyzed in Yan et al. (2019), allowing for a direct comparison between different experimental setups, including boundary layer and tropopause source domains, as well as ten years and one year of simulated duration.

## 3 Air mass fractions from the Asian monsoon boundary layer

In this section, we quantify the air mass fractions transported from the boundary layer of the source domains. Our investigation explores the transport of air originating from all the source regions to the global stratosphere, utilizing artificial tracers released at the surface layer. The transport contributions from both the NH extratropics and SH extratropics are small, each accounting for less than 0.5%. Consequently, we will not present detailed discussions regarding the results from these two source regions in this study.

**3.1 Transport contributions from boundary-layer source regions to the global stratosphere**

To assess the global contributions originating from the AM region, we calculate zonally averaged seasonal mean AMFs from the AM boundary layer (Fig. 1, left panels). The transport characteristics from the subTR-NH exhibit significant similarities to those from the AM region; therefore, we do not present results for the subTR-NH in this study. In contrast, we illustrate the AMFs from the same latitude band as for the AM region but in the Southern Hemisphere (subTR-SH) (Fig. 1, middle)

to explore hemispheric differences. Additionally, for reference, we include the AMFs from the tropics (Fig. 1, right). Despite the fact that the AM region is approximately four times smaller than the subTR-SH, the contributions of air masses from the AM region to the global stratosphere, including the southern hemispheric stratosphere, are approximately 1.5 times larger than the corresponding contributions from the subTR-SH. However, it is important to note that the AMFs originating from the AM region to the global stratosphere are approximately 2-3 times smaller than those originating from the tropics.

The transport of newly released AM air into the NH stratosphere initiates during boreal summer, such that its impact on the NH stratosphere maximizes during boreal autumn. The AM origin tracers are also transported to the tropical pipe and the SH in the following seasons. Transport patterns from the subTR-SH to the stratosphere exhibit a lot of similarities to those originating from the AM, albeit with a 6-month time shift. It is noteworthy that more than 50% of the stratospheric air is transported from the tropics, representing the largest contribution to the global stratosphere among all the source regions. The AM air tends to

accumulate in the polar middle stratosphere, particularly above approximately 450 K. In the tropics, the fraction is lower due to wintertime air ascending; however, in high latitudes, there is evidently a surplus of summertime air compared to wintertime air making its way into this region.

We use the zonally averaged meridional wind to examine the dynamic factors influencing transport from the source regions. During the boreal summer (JJA), the jet streams in the NH and SH exhibit differing characteristics, with the NH jet being



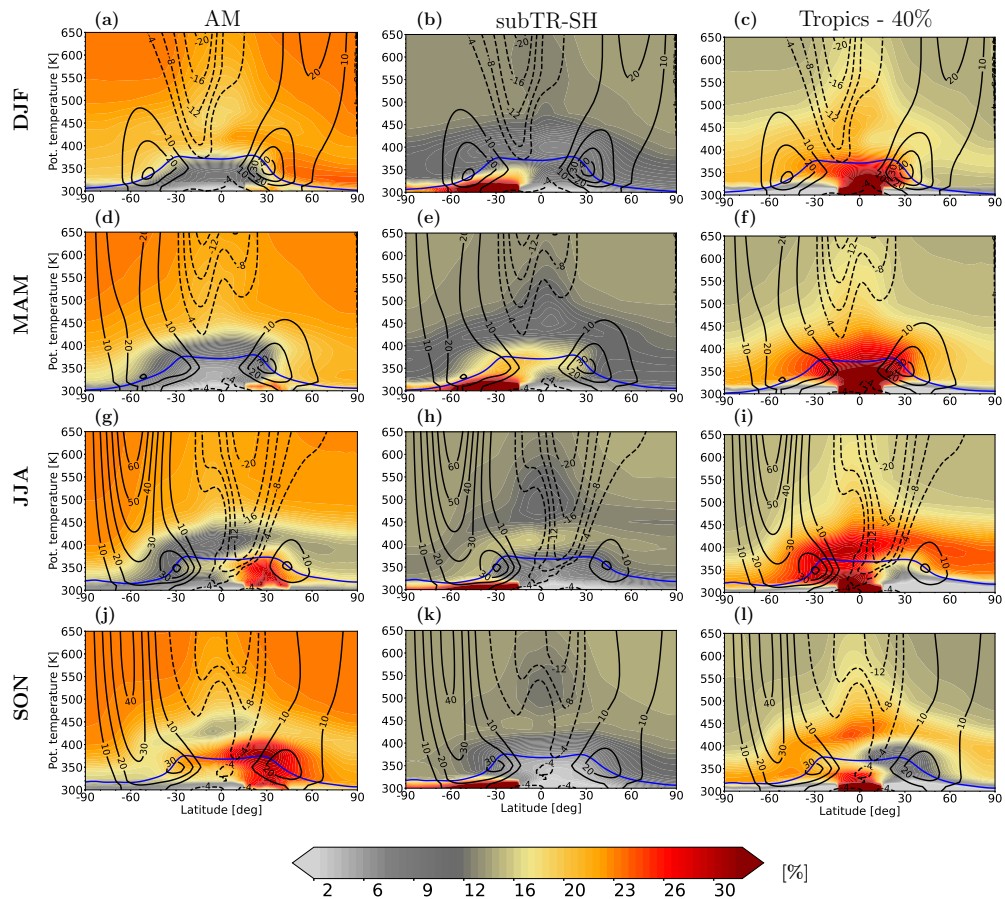

**Figure 1.** Climatological (2010–2013) zonal mean wind (black contours) and air mass fractions (color shading) originating from the AM (15-45° N, 30-120° E, left), SH subtropics (15-45° S, middle), and tropics (15° N-15° S, right) for different seasons (rows). The blue line shows the WMO tropopause. Note a 40% mass fraction has been subtracted from the tropical mass fraction for plotting.

relatively weak and the SH jet being notably strong. During the subsequent seasons of SON and DJF, the tropical easterlies and SH jets tend to weaken, while the NH jet intensifies. The strength of the polar vortex plays a pivotal role in governing the transport of newly released tracers. A weak polar vortex facilitates the transport of tracers into the polar stratosphere, while a strong polar vortex effectively isolates stratospheric air within the polar regions. Notably, during DJF, the polar vortices in the NH and SH respectively reach their strongest and weakest phases. The strong Pacific westerly ducts during boreal autumn and winter enable large cross-hemispheric transport (see Yan et al., 2019).

To explore the seasonal variation of transport from the boundary layer, we remove the annual mean contributions from each source region. These seasonal anomalies of the climatological zonal mean AMFs are depicted in Fig. 2, showcasing data from the AM region (left), subTR-SH (middle), and the tropics (right). While the absolute contributions from the AM, subTR-SH, and tropics differ significantly, the amplitude of the AMF seasonal cycle in the global UTLS, originating from these three



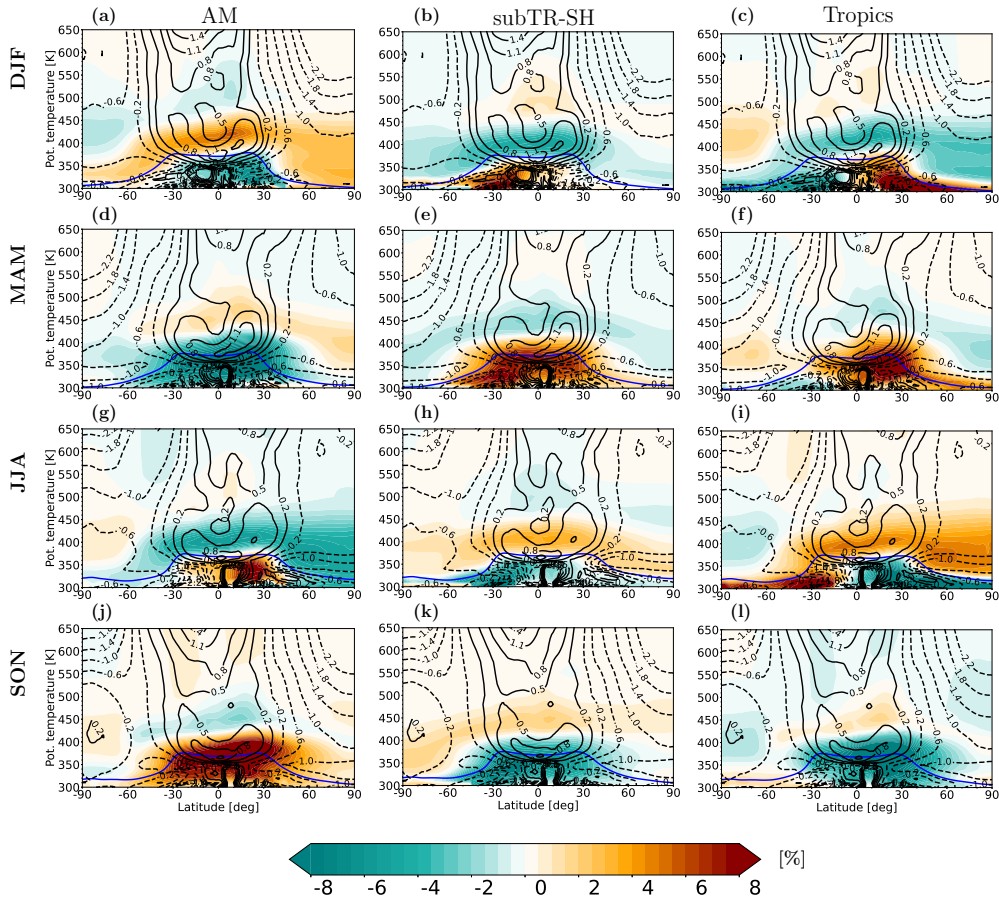

**Figure 2.** Climatological (2010–2013) air mass fraction (AMF) anomaly displayed as the departure from the annual average (color shading), overlaid with ERA5 zonal mean total diabatic heating rates (black contours). The thick blue line represents the WMO tropopause.

source regions, demonstrates comparable values. Evidently, newly released AM air primarily undergoes transport from the troposphere to the stratosphere during boreal summer and autumn. During boreal summer, air of AM origin is more confined to the NH. However, as boreal autumn sets in, these air masses undergo inter-hemispheric transport, particularly below the tropical tropopause, which may be associated with the Hadley circulation. In the following seasons, AM origin air is transported into the SH extratropics through the UTLS, driven by the Brewer-Dobson circulation. Transport patterns from the subTR-SH to the stratosphere exhibit considerable similarities to those from the AM region. It is noteworthy that subTR-SH air mass anomalies portray entirely negative values in the lower stratosphere above the SH extratropics during the months of MAM (Fig. 2e). In contrast, transport patterns from the tropics demonstrate nearly the opposite sign compared to those from the AM region. Tropical and AM origin air tend to compensate for each other within the atmosphere. Combined, the total tracer contributions from these two source regions amount to approximately 80% of the AMFs in the global atmosphere.



In Fig. 2, the total diabatic heating rate (black contours) reveals pronounced downward motion in the northern polar strato-
sphere during SON and DJF. This downward motion effectively isolates a significant portion of the stratospheric air originating
from the AM region within the Arctic polar vortex (Fig. 2a and Fig. 2j). During JJA, the Arctic stratosphere experiences its
weakest westerly jet (Fig. 1i), allowing for the substantial transport of newly released tropical air into the Arctic stratosphere
(Fig. 2i). This suggests that transport contributions from the tropics to the Arctic stratosphere are particularly influenced by
quasi-horizontal isentropic transport.

Notably, the seasonality of the total diabatic heating rate over the Antarctic region exhibits a six-month shift compared to
that over the Arctic. In the Antarctic region, strong downwelling occurs during MAM and JJA, leading to the accumulation of
significant AM origin tracers (Fig. 2g). The largest amount of tropical tracers is transported into the Antarctic during DJF due
to the weakened jet barrier and downwelling over the SH extratropics ((Fig. 1c and Fig. 2c). In summary, more AM tracers are
transported to the polar stratosphere during local autumn and winter, while a greater quantity of tropical tracers is transported
to the polar stratosphere during local summer.

### 3.2   Transport time scales to the polar stratosphere as inferred from the age of air spectrum

In this subsection, we investigate the transport from the source regions into the polar regions specifically. Therefore, we analyze
age spectra along the latitudes of 60° N and 60° S and present the results in Fig. 3 and Fig. 4, respectively. Examining the age
spectra of air originating from the AM, subTR-SH, and tropics along the 60° N latitude (Fig. 3) reveals a clear seasonality in
transport into the Arctic region, particularly for air from the AM and subTR-SH source regions. Below about 450 K, transport
from the boundary layer of the AM, subTR-SH, and tropics to the Arctic region occurs mainly on timescales smaller than about
2 years. Above 450 K, on the other hand, transport timescales to the northern polar vortex are general larger than 2 years. The
separation in the age spectra below/above 450 K and transit times shorter/longer than 2 years might indicate the separation
between shallow and deep branches of Brewer-Dobson circulation. Notably, the air from the three source regions exhibits its
youngest mean age of air (AoA) in SON over the Arctic.

The age spectrum of air originating from the AM region exhibits large values when the tracers are released during boreal
summer (JJA), hence the probability of transport to the polar lower stratosphere maximizes for air masses released at the
surface during summer. Conversely, these values are nearly zero when the AM origin tracers are released during boreal winter
(DJF), strongly suggesting that pollutants from the AM region are primarily transported to the Arctic stratosphere during boreal
summer and autumn. In contrast, the primary contributions from the subTR-SH and tropics to the Arctic occur during austral
summer (DJF). The age spectrum of AM origin air shows larger partial contribution to the Arctic compared to that from the
subTR-SH, both of them are evidently smaller than that originating from the tropics. Additionally, the transport patterns of
subTR-SH and tropical origin air exhibit a lot of similarities, with a six-month shift compared to the AM origin air. Transit
times from the AM to the Arctic are longer (shorter) than those from the subTR-SH and tropics during JJA (DJF).

The age spectra along 60° S, as presented in Fig. 4, exhibit numerous similarities to those observed along 60° N. Specifically,
the PDFs display substantial values below (above) the lower stratosphere, corresponding to transit times shorter (longer) than 2
years. The age spectra from both 60° N and 60° S show the different impacts of deep and shallow branches of Brewer-Dobson



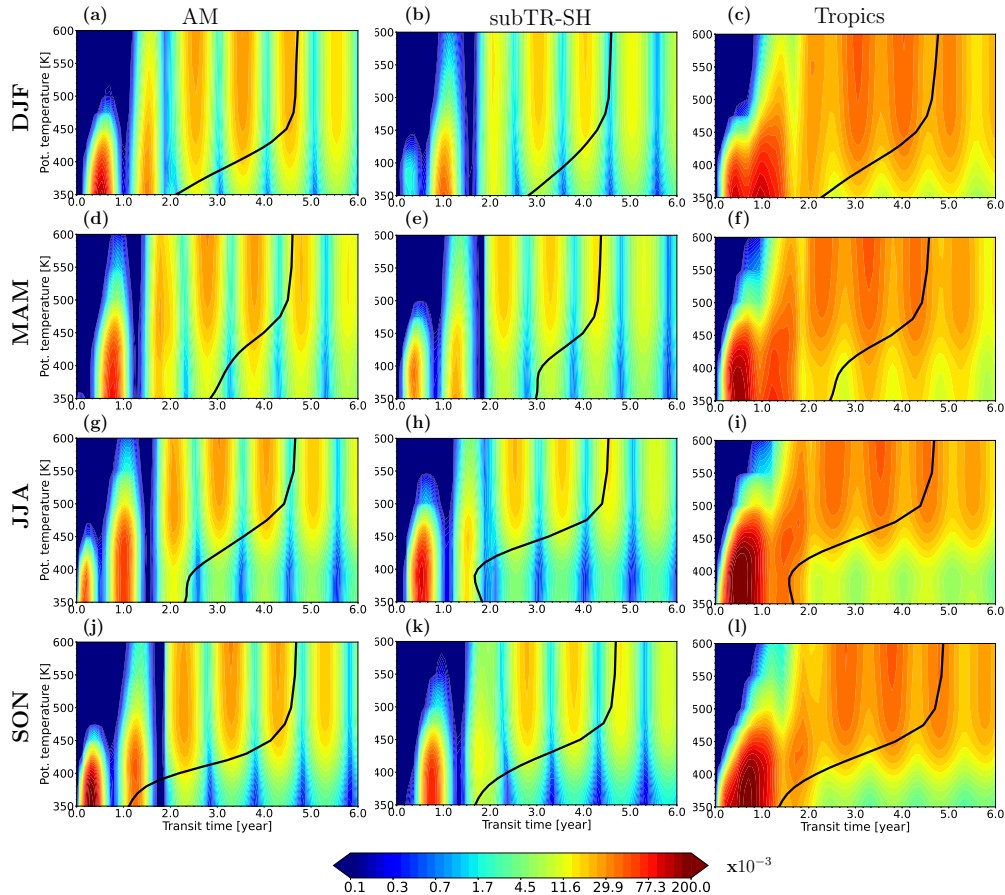

**Figure 3.** Age spectra as (partial) transit time probability density functions (PDFs), calculated for air originating from three source regions: AM, SH subtropics, and tropics. Age spectra are shown along 60° N for all seasons. The black line shows the mean AoA as derived from the age spectrum.

circulation on the transport of air mass into the polar regions. The age spectra along 60° S also show distinct seasonal cycle
features. The PDFs along this latitude reveal peaks and nadirs at transit times similar to those along 60° N. This recurring
pattern underscores the fact that pollutants from the source regions released during summer have the highest chance to be
transported to the Antarctic stratosphere while pollutants released during the winter face greater difficulty in distant transport.

Despite the similarities, it is noteworthy that the values of the PDFs along the 60° S latitude are notably smaller when
compared to those along 60° N. This fact holds true even for the subTR-SH origin air, despite the shorter distance between
subTR-SH and the Antarctic region compared to the distance between subTR-SH and the Arctic. The variation in the vertical
range of source tracers transported to the Antarctic stratosphere is likely related to the strength of the polar vortex. During
austral winter (JJA), the source tracers experience a downward transport to lower levels over the Antarctic region compared
to austral summer (DJF). Additionally, both the vertical range and vertical profile of the mean AoA originating from the three





source regions along the 60° S latitude exhibit a shift of approximately 6 months when compared to those observed along

60° N. Consequently, air from these three source regions over the Antarctic region exhibits its youngest mean AoA in MAM.

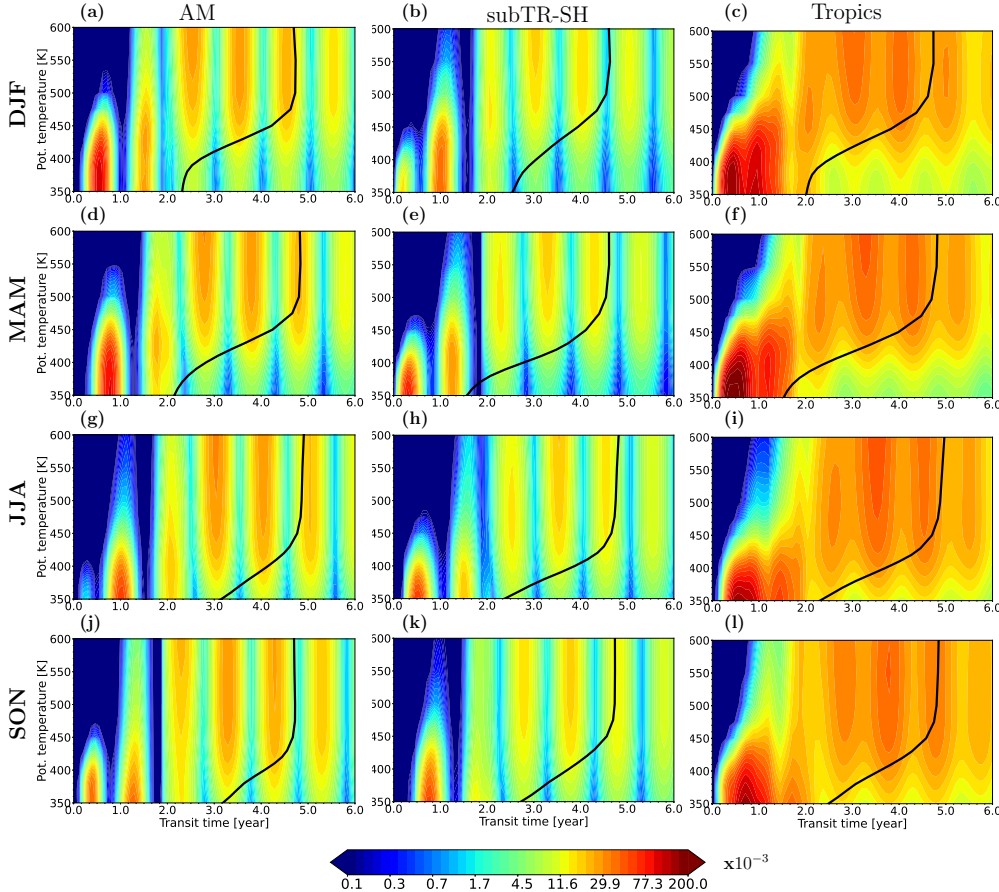

**Figure 4.** Same as Fig. 3 but for age spectra along 60° S.

### 3.3    Transport time scales to the polar lower stratosphere driven by the shallow branch of Brewer-Dobson circulation

The notable difference in age spectrum behavior at different altitudes for transit times shorter and longer than 2 years has motivated our investigation into the contributions of source tracers transported to the polar regions with transit times less than 2 years, potentially driven by the shallow branch of the Brewer-Dobson circulation. In this way, our focus is directed more

towards the short-lived boundary emissions transported into the polar regions, influenced more by fast isentropic transport and less by the upper branch of the Brewer-Dobson circulation. In Fig. 5, we present AMFs and their anomalies with AoA less than 2 years, which are transported to the northern polar region [60-90° N] and the southern polar region [60-90° S]. Young air, characterized by an AoA of less than 2 years, originating from all three source domains is primarily transported





to the polar regions below the polar vortex, and this transport decreases with increasing altitude (Fig. 5a-f). Interestingly,
despite the smaller size of the AM region and the closer distance between the subTR-SH and the Antarctic region, the air mass
contributions from the AM to the polar regions are even larger than those from the subTR-SH. However, both AM and subTR-
SH contributions remain much smaller than those from the tropics. Furthermore, the tracers from all three source regions exhibit
a greater transport into the northern polar region when compared to their transport into the southern polar region. Maxima and
minima AMFs from the three source regions over [60-90° N] exhibit a 6-month shift when compared to those over [60-90° S].
Specifically, air originating from the AM and subTR-SH regions displays peaks in the northern and southern polar regions
during local autumn, while air from the tropics reaches its peak in the polar region during local summer.

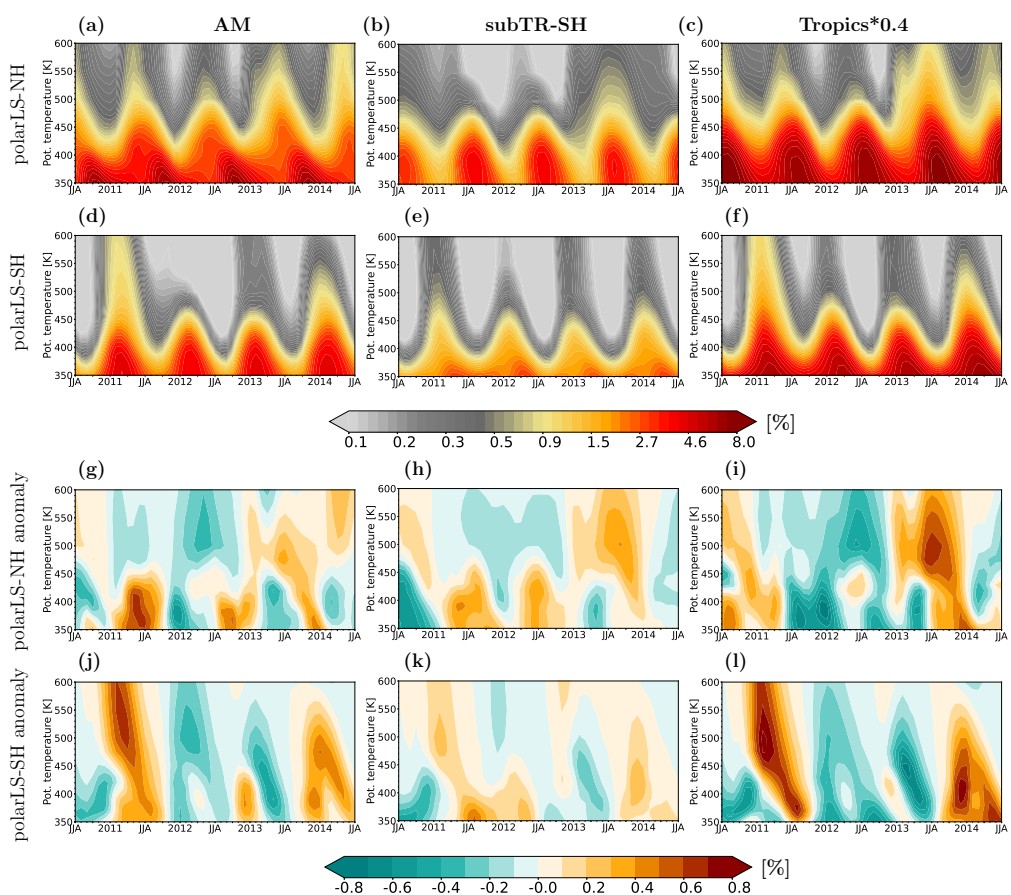

**Figure 5.** Potential temperature–time cross sections of young (< 2 years) air mass fractions and anomalies relative to the monthly mean value
(2011-2014) originated from AM (left), SH subtropics (middle), and tropics (right) over 60° N−90° N and 60° S−90° S.

We perform deseasonalization on the AMFs transported from the source regions to the polar regions and present the results
in Fig. 5g-l. Notably, there exists distinct inter-annual variability in the AMFs over the polar regions, and the magnitude of
this variability is comparable between the northern and southern polar regions. These inter-annual variability patterns vary





significantly across different vertical levels. The tracers transported to the polar regions display less variability above 450 K than below. The inter-annual variability of AMFs over the southern polar region exhibits a 6-month shift compared to that of the northern polar region. More frequent changes of the inter-annual variability below 450 K over the polar regions compared to the higher levels are likely driven by both downward propagation from higher altitudes and the shallow branch of the Brewer-Dobson circulation.

## 4    Quantification of Asian monsoon impact on the polar stratospheric composition


Up to this point, we have calculated the contributions from the AM boundary surface layer and the other two source regions to the global stratosphere. Our investigation of the transport into the polar regions has revealed that long-term transport contributions from the AM to the polar regions fall within the range of 5% to 20%, depending on the altitude. It is worth noting that the AMFs originating from the AM boundary surface layer over the polar region below 450 K are predominantly composed of
air with transit times of less than 2 years. In this section, our objective is first to validate and then to quantify the impact of emissions from the AM boundary as well as other source regions on pollutant concentrations over the polar regions. We achieve this using $SF_6$ observations obtained from the ACE-FTS satellite instrument and reconstructed $SF_6$ data by using Equation 1.

### 4.1    Evaluation of reconstructed $SF_6$ data

We utilize the AMFs originating from different source regions and zonally averaged $SF_6$ data from NOAA surface observa-
tions to reconstruct zonal mean $SF_6$ data in the whole atmosphere. Here, the original NOAA $SF_6$ surface observation data are used to estimate the mean transit time from station to station. Measurements of $SF_6$ are offset for each latitude by the corresponding transit time to get high resolution latitudinal data. The reconstructed $SF_6$ data is calculated by averaging $SF_6$ surface observations from each source region, integrated with the corresponding AMFs originating from that same source region. To assess the accuracy of the reconstructed $SF_6$ data, our first step involves a comparison between the reconstructed data and $SF_6$
observations obtained from ACE-FTS. In Fig. 6a-d, we present the seasonal mean $SF_6$ concentrations from ACE-FTS (left) and the reconstructed data (right) for both the north polar region (a-b) and the south polar region (c-d). The comparison results reveal a strong agreement between the vertical distribution of the reconstructed $SF_6$ data and the ACE-FTS observations over the polar regions. The reconstructed $SF_6$ data is slightly smaller in the troposphere and larger in the stratosphere compared to ACE-FTS $SF_6$ data. This is due to underrepresented CLaMS vertical transport in the troposphere, truncation of calculated age
spectra to 10 years, and missing mesospheric $SF_6$ loss in the reconstruction. Nevertheless, both data sets exhibit rapid growth rates and distinct seasonality. Specifically, $SF_6$ concentrations over the northern polar region peak in SON, while those over the southern polar region peak in MAM. Furthermore, the abundance of $SF_6$ over the northern polar region is larger than that over the southern polar region.

To assess the reliability of the reconstructed $SF_6$ data, we implement a process of detrending and deseasonalizing $SF_6$
distributions over the polar regions. Figure 6e-h present the detrended and deseasonalized $SF_6$ data from both ACE-FTS and the reconstruction for both the north polar region and the south polar region. Upon examining the inter-annual variability over





**Figure 6.** Potential temperature–time of seasonal mean SF$_6$ (a-d) of ACE-FTS (left) and reconstruction (right) over the north polar region (60° N−90° N) and the south polar region (60° S−90° S) and their detrended and deseasonalized distribution (e-h).

the polar regions without trends, the comparison shows clear similarities between the reconstructed SF$_6$ and ACE-FTS SF$_6$, and both of them also show similar features to the inter-annual variability of the three source origin AMFs over the polar regions (refer to Fig. 5g-i). Notably, the detrended and deseasonalized SF$_6$ data for the northern polar region exhibit a clear positive





anomaly during JJA in 2010 and 2013, particularly above 450 K. This positive anomaly subsequently propagates downward and leads to a positive anomaly in DJF of 2010 and 2013 below 450 K. Similar positive anomalies below about 450 K also occur in the JJA of 2011 and 2012, as the patterns observed for AM and subTR-SH origin air (see Fig. 5g and Fig. 5h).

In the southern hemisphere, both the reconstructed $SF_6$ and ACE-FTS $SF_6$ observations reveal inter-annual variability characterized by positive anomalies in 2011 and 2013. However, the positive anomaly observed in the reconstructed $SF_6$ data for 270 2012 does not align with the inter-annual variability in ACE-FTS $SF_6$ observations between 400 K and 550 K. This discrepancy warrants further investigation. Overall, the inter-annual variability in the reconstructed $SF_6$ data without trends closely matches that observed in ACE-FTS $SF_6$ data over the polar regions. The reconstructed $SF_6$ data successfully captures most of the key features observed in ACE-FTS $SF_6$ data over the polar regions.

## 4.2 Contributions of $SF_6$ transport from source regions

The reconstructed $SF_6$ data have been validated with ACE-FTS $SF_6$ observations above. We now turn our attention to analyzing the contributions of transport from each source region to the polar regions using the reconstructed $SF_6$ data. Figure 7 provides a comprehensive view, showcasing the reconstructed $SF_6$ concentrations alongside the relative contributions from AM, subTR-NH, subTR-SH, and TR for both the northern polar region (left) and the southern polar region (right). In the northern polar region, the $SF_6$ levels originating from the source regions exhibit an increasing trend from 2010 to 2014, with the exception of 280 transport from TR in 2010. These concentrations display distinct seasonality within the lower stratosphere of the polar region. The transport contributions from AM and subTR-NH peak during DJF, while those from subTR-SH and TR reach their crest values in JJA. This might be related to the transit time in summer from source regions to the polar lower stratosphere, and pollutants released during the summer are more extensively transported to the polar regions compared to the winter emissions of pollutants from source regions as we discussed above in Sect. 3.2. Interestingly, despite increasing global boundary layer 285 emissions from 2010 to 2014, the $SF_6$ concentrations transported from AM, subTR-NH, and subTR-SH to the polar region do not consistently exhibit an upward trend. This suggests that factors beyond emissions might exert a substantial influence on pollutant abundance in the southern polar region, such as the dynamics of the polar vortex and the Brewer-Dobson circulation.

We normalize the $SF_6$ emissions from each source region by the total $SF_6$ from the global surface layer, which allows us to determine the relative contributions of $SF_6$ transport from each source region to the polar regions. Notably, the relative 290 contributions of $SF_6$ from AM, subTR-NH, and subTR-SH to the polar regions above 450 K are greater than those to the regions below 450 K, while the tropics contribute more to altitudes below 450 K in the polar regions. Specifically, AM contributes over 20% of the $SF_6$ present in the polar stratosphere above 450 K, and remarkably, more than 70% of the subTR-NH $SF_6$ in the polar region originates from AM transport. It is worth noting that the reconstructed $SF_6$ from AM region is based on the zonally averaged $SF_6$ data from NOAA surface observations at the same latitude bands, which might even underestimate the 295 contributions of $SF_6$ from AM region. Moreover, the contributions of $SF_6$ from AM to the southern polar region slightly surpass those to the northern polar region, underscoring the significance of inter-hemispheric transport dynamics. Again, the $SF_6$ contributions from AM outstrip those from subTR-SH despite the smaller source domain of AM region and the longer distance between AM and the southern polar region. Another interesting fact is that the positive $SF_6$ anomaly in the reconstruction over





**Figure 7.** Potential temperature–time sections of reconstructed SF$_6$ (black contours) and relative contributions (color shading) from AM (a and b), NH subtropics (c and d), SH subtropics (e and f), and tropics (g and h) over the north polar region (60° N−90° N, left) and the south polar region (60° S−90° S, right). Note that the relative contributions have been scaled for plotting.

the southern polar region in 2012 (Fig. 6h) is related to particularly strong transport from AM region (Fig. 7b). The tropics

300   still serve as the largest SF$_6$ source for the polar regions, accounting for over 50% of SF$_6$ concentrations in the polar regions.




Considering the size of their respective source domains, the transport of SF$_6$ from the AM boundary layer to the polar region proves to be more efficient compared to that from the tropical boundary layer.

## 5   Discussion

The transport of pollutants from South-East Asia into the global stratosphere has attracted significant attention due to the rapid
economic growth in this region and the unique pathway of transport through the ASM anticyclone. The quantification of this transport can be influenced by various methods, models, and reanalysis datasets. Previous studies by Garny and Randel (2016) and Yu et al. (2017) investigated the transport from the upper troposphere over the AM region, employing trajectory analyses and aerosol simulations, respectively. Ploeger et al. (2017) conducted an assessment of pollution transport from the ASM into the lower stratosphere using artificial tracers of air mass origin from the ASM anticyclone. Although the simulation periods
and source domain sizes slightly differed across these studies, they collectively revealed that approximately 15% of air parcels originating from the AM region are directed toward the extratropics of the NH, with less than 5% reaching the extratropics of the SH. Furthermore, prior investigations based on one-year simulations also indicated that air of AM origin experiences peak transport to the northern polar lower stratosphere during September to November (e.g. Ploeger et al., 2017; Yu et al., 2017) and to the southern polar lower stratosphere from December to February (e.g. Yan et al., 2019). These findings align closely with
the results obtained in the present study, which is based on a ten-year simulation.

One notable difference between our study and prior research lies in the inter-hemispheric contrast in transport contributions from the AM region to the polar stratosphere above 450 K. Our analysis reveals that the contributions from the AM to the southern polar stratosphere and the northern polar stratosphere are roughly equivalent, both exceeding 15% (Fig. 7). These new findings are mainly attributed to simulation times spanning a decade, significantly longer than in previous studies where
simulations over one year or shorter were presented. In these studies, AM origin air was primarily transported toward the NH, with limited penetration into the higher altitudes above 450 K in the southern polar region. Our findings, in contrast, reveal a more complex pattern: AM air tends to be transported to altitudes below 450 K with transit time less than 2 years, whereas for longer transit times, AM origin air reaches the high altitude above 450 K in the polar stratosphere via the upper branch of the Brewer-Dobson circulation.

Another source of differences between the previous studies and the findings discussed here originates from the differences in the vertical position of the domain emitting the AM origin tracer. In this work, we set the source domain in the boundary layer of the AM region, while Yan et al. (2019) conducted simulations of transport from 350−360 K and 370−380 K over the AM region to the global stratosphere using ERA-Interim and Merra-2 reanalysis meteorology. Notably, Garny and Randel (2016) simulated trajectories from 360 K, but their simulation duration of just 60 days was insufficient to facilitate significant transport
of AM origin air into the southern polar stratosphere. The domain considered in Yu et al. (2017) was situated immediately below the tropopause leading to high transport contributions to the NH compared to those to the SH. Collectively, these prior studies suggest that when air is close to the tropopause, AM origin air tends to be retained more in the NH, potentially underestimating its impact on the SH, particularly in the southern polar region.



Another open question pertains to the factors contributing to the inter-annual variability of transport from source regions
to the polar stratosphere. One possible influence could be the Quasi-Biennial Oscillation (QBO). The easterly phase of the
Quasi-Biennial Oscillation (QBO) in 2010 and 2013 (e.g. Anstey et al., 2022) corresponded with positive anomalies in 2010
and 2013 in the northern polar stratosphere and in 2011 and 2014 in the southern polar stratosphere. The easterly phase of
the QBO induces positive upwelling anomalies near the equator, as previously demonstrated by Baldwin et al. (e.g. 2001);
Brown et al. (e.g. 2023), and also influences the strength of the stratospheric polar vortex. In general, the polar vortex tends to
be weaker during the easterly QBO phase when compared to the westerly phase (e.g. Holton and Tan, 1980; Pascoe et al., 2005;
Watson and Gray, 2014). Therefore, the combination of stronger tropical upwelling and a weaker polar vortex transport barrier
during QBO easterly phases could cause positive anomalies in $SF_6$ at polar latitudes. Another potential contributing mechanism
could involve variations in tropical sea surface temperatures and the strength of Asian monsoon anticyclone. Comprehensive,
long-term simulations and further research efforts will be essential for gaining a deeper understanding of the mechanisms
responsible for driving the inter-annual variability in transport into the polar stratosphere.

## 6 Conclusions

In this paper, we first quantify the air mass transport contributions from the AM region to the polar stratosphere. This quantifica-
tion is achieved using artificial tracers initialized at the boundary layer within CLaMS simulations driven by ERA-5 reanalysis
data. Additionally, we incorporate transport from the subTR-SH to explore hemispheric differences, while transport from the
tropics serves as a reference point. Notably, despite the size of AM region being approximately four times smaller than that
of subTR-SH, our results reveal that air mass contributions from the AM to the global stratosphere, including the southern
hemispheric stratosphere, are approximately 1.5 times larger than the corresponding contributions from the subTR-SH. How-
ever, it is worth noting that transport from the tropical boundary layer still predominates in the stratosphere. Specifically, the
contributions of transport from the tropical boundary layer to the global stratosphere are roughly double those originating from
the AM boundary layer. Additionally, strong downwelling and jet streams in the polar stratosphere play a vital role in isolating
stratospheric air masses within the polar vortex during local autumn and winter. Consequently, this phenomenon leads to an
increase in AM tracers in the polar stratosphere during these seasons. On the other hand, weak jet streams during local summer
allow the isentropic transport of tropical tracers into the polar lower stratosphere.

The age spectrum analysis highlights that transport from the AM boundary layer to the polar vortex above 450 K primarily
occurs at timescales exceeding 2 years, whereas transport timescales to the polar regions below the vortex are typically shorter,
falling below the 2-year threshold. Our examination segregates transport contributions into two categories: those occurring
within two years and the total contributions over the long term. This distinction reveals a notable presence of inter-annual
variability in the tracer transport originating from the source domain to the polar regions. This inter-annual variability in air
mass composition above the polar vortex has the potential to propagate downward, and affects also the region below the vortex.
An additional factor contributing to this inter-annual variability is the influence of the shallow branch of the Brewer-Dobson
circulation.

Furthermore, we employ the reconstruction of $SF_6$ to provide further insights into the extent of transport contributions originating from the AM region. To validate the accuracy of our reconstructed $SF_6$ data, we compare it with $SF_6$ observations obtained from the ACE-FTS satellite instrument. The results demonstrate that the AM region plays a substantial role, con-
370 tributing over 20% of the $SF_6$ content in the polar stratosphere at altitudes above 450 K. This contribution even surpasses that of the subTR-SH, despite the comparatively smaller geographical expanse of the AM domain. Impressively, more than 70% of the subTR-NH $SF_6$ found in the polar regions can be attributed to transport originating in the AM region. While tropical origin air exerts the most significant influence on the composition of the polar stratosphere, our findings highlight the superior efficiency of transport from the AM boundary layer to the polar region and potentially significant influence of ozone-depleting
375 substances from South-East Asia on polar stratospheric chemistry, especially when accounting for the different sizes of the respective source domains.

*Data availability.* The $SF_6$ data are available at the website https://gml.noaa.gov/aftp/data/hats/sf6/combined/GML_global_SF6.txt (last access: 15 March 2024). The CLaMS model outputs may be obtained from the authors upon request.

*Author contributions.* XY conducted the ERA5-driven model simulations and performed the data analysis. PK, FP, and AP played key roles
380 in designing the analysis, offering valuable insights through discussions, and providing constructive feedback. XY wrote the paper with contributions from all co-authors.

*Competing interests.* Some authors are members of the editorial board of journal ACP.

*Acknowledgements.* We thank the ECMWF for providing ERA-Interim meteorological reanalysis data for this study. We gratefully acknowledge the computing time for the CLaMS simulations granted through VSR project ID JICG11 on the supercomputer JURECA
385 at Forschungszentrum Jülich. This research has been supported by the National Natural Science Foundation of China project (grant no. 42275093 and 41905040)).



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
