# Peer review of "Transport into the polar stratosphere from the Asian monsoon region"

_EGUsphere, 2024_

## Author Response (AR1)

**Referee1**

Many thanks to the reviewer for the comments, they have helped to improve the clarity of the manuscript. In the following, we address all the points raised in the review (denoted by italic letters). Text changes in the manuscript are highlighted in red or blue.

**General comments**

*The paper presents an analysis of the contribution of air masses originating in the Asian monsoon boundary layer region to the polar stratosphere using ClaMS model with ERA-5 trajectories. The results show a relevant role of the AM, which contributes to approximately 20% of the air mass in the polar stratosphere in both hemispheres. The major contribution is of air from the tropics, consistent with upwelling year-round in that region. The results are novel and I suggest publication after addressing the following minor comments requesting clarification on the interpretation of some aspects.*

**Specific comments**

(1) **L91-92: if the boundary condition is maintained during 30 years this would not be a delta function, but rather a step function?**

A. Inside the source regions, the boundary condition is always a delta function with a width of 30 days in the layer of 0-100 K in CLaMS zeta coordinates (about the lowest 1.5 km, orography-following, approximating the boundary layer). The pulse tracers are maintained for 10 years outside of the source regions, then reset to 0 everywhere (including the source region).

(2) **L122-123: does this 0.5% refer to the contribution to the polar regions? I would think that substantial transport from midlatitudes to poles occurs within the troposphere. It would be beneficial to discuss the limitations in the tropospheric transport representation.**

A. Here 0.5% refers to the air mass fractions transported from the NH extratropics and SH extratropics to the global stratosphere. The NH extratropics and SH extratropics are defined as the area spanning 45-90° N and 45-90° S, respectively. Figure RL 1 and Figure RL 2 show the air mass fractions from the boundary layer of NH extratropics and SH extratropics. The contributions from these two source regions to the global stratosphere are less than 0.5% except the lowermost stratosphere (below 330 K) over the polar stratosphere. The main focus of this study is to understand the transport from the Asian monsoon region to the polar regions. Other source regions with large contributions are included as a reference. Hence we didn't include the transport contributions from the NH extratropics and SH extratropics to the global stratosphere. Indeed the transport contributions from the NH extratropics and SH extratropics to the polar troposphere are substantial. However, more chemical processes should be included to quantify the transport contributions into the troposphere in the simulations.

(3) **L143: 'We use zonally averaged meridional wind' $\longrightarrow$ zonal wind**

A. It is changed in the manuscript.

(4) **L141-142: 'In the tropics...' Why are you referring to wintertime and summertime air here? The**

[Figure]

Figure RL 1: Climatological (2010–2013) zonal mean air mass fractions (color shading) originating from the NH extratropics (45-90° N) for different seasons.The black line shows the WMO tropopause.

*tropical ascent does not only happen in wintertime, and I do not understand the 'surplus of summertime air' in high latitudes*

A. In the tropics, here means that the AM origin air mass fraction during wintertime is lower than that during other seasons due to relatively strong ascending movement in the tropics during wintertime, which leads to larger contribution from tropics during wintertime.

A lot of AM origin air is transported to the high latitude region during summer driven by the Asian summer monsoon circulation, while the AM origin air is more suppressed and isolated inside the source region during winter. Hence there is evidently a surplus of summertime air compared to wintertime air making its way into the high latitude regions.

(5) *L149-150: 'The strong Pacific westerly ducts during boreal autumn and winter enable large cross-hemispheric transport (see Yan et al., 2019).' In which features do you see this? According to the patterns in figures 1 and 2, cross-hemispheric transport of AM air into the SH peaks in SON near the tropopause, but cross-hemispheric transport of subTR-SH into the NH peaks in MAM, also near the tropopause.*

A. The reference cited in the manuscript was wrong. It should be Yan et al.(2021), which is changed in the manuscript. Figure 1j shows that large amount of AM air is transported cross the equator near the tropopause. Here we just show the zonally mean results, hence we can not see the influence of the westerly ducts directly. Yan et al.(2021) reported that the westerly ducts over the Pacific allow cross-hemispheric transport of air from the NH to the SH near the tropopause. This mechanism does not explain the cross-

[Figure]

Figure RL 2: Same as Figure RL 1 but for air mass fractions originating from the SH extratropics (45-90° S).

hemispheric transport from the SH to the NH.

Yan, X., Konopka, P., Hauck, M., Podglajen, A., and Ploeger, F.: Asymmetry and pathways of inter-hemispheric transport in the upper troposphere and lower stratosphere, Atmospheric Chemistry and Physics, 21, 6627–6645, https://doi.org/10.5194/acp-21-6627-2021, 2021.

(6) *L155-156: 'Evidently, newly released AM air primarily undergoes transport from the troposphere to the stratosphere during boreal summer and autumn'. Maybe this could be rephrased to avoid confusion. The positive anomalies of AM air are limited to the troposphere in JJA, and seen in the stratosphere in SON and DJF. In general, the results suggest that the AM air remains confined during JJA and only reaches the extratropical stratosphere in SON.*

A. It is rephrased as "Evidently, newly released AM air is primarily limited to the troposphere during boreal, transported deeply to the stratosphere during boreal autumn and winter." in the manuscript.

(7) *L165-169: In this paragraph it is unclear whether you are referring to the upper levels or the lower levels, please clarify.*

A. This paragraph as "In Fig.2, the total diabatic heating rate (black contours) reveals pronounced downward motion in the northern polar stratosphere during SON and DJF. This downward motion effectively isolates a significant portion of the stratospheric air originating from the AM region within the Arctic polar vortex above 450 K (Fig.2a and Fig.2j). During JJA, the Arctic stratosphere experiences its weakest westerly jet (Fig.1i), allowing for the substantial transport of newly released tropical air into the Arctic stratosphere between tropopause and 450 K (Fig.2i). This suggests that transport contributions from the

tropics to the Arctic lower stratosphere are particularly influenced by quasi-horizontal isentropic transport." in the manuscript.

(8) **_L171: 'Notably, the seasonality of the total diabatic heating rate over the Antarctic region exhibits a six-month shift compared to that over the Arctic.' Not really a 6-month shift, MAM downwelling is larger than JJA in the SH, while in the NH it peaks in DJF._**

A. We agree it is not absolute six-month shift. The downwelling in DJF is slightly stronger than that in SON above 450 K over the arctic, while it is opposite below 450 K. The downwelling in both DJF and SON are stronger than those in MAM and JJA. For the Antarctic, the downwelling in MAM is stronger than that in JJA in the whole stratosphere. The downwelling in both MAM and JJA are also stronger than those in SON and DJF over the Antarctic. We changed the sentence as 'Notably, the seasonality of the total diabatic heating rate over the Antarctic region nearly exhibits a six-month shift compared to that over the Arctic.' in the manuscript.

(9) **_L185-186: 'Notably, the air from the three source regions exhibits its youngest mean age of air (AoA) in SON over the Arctic.' For tropical air it is actually JJA, no? In general throughout Section 3.2, it is unclear if you are referring to levels above/below 450 K, or average of all levels?_**

A. Thank you for the comment. Indeed, the description is not accurate. Here, it refers to the levels below 450 K. This information is now included in the manuscript. We have went through this section and clarify the conclusions about the figures.

(10) **_L194-195: 'Transit times from the AM to the Arctic are longer (shorter) than those from the subTR-SH and tropics during JJA (DJF).' How can this be? I understand having efficient inter-hemispheric transport on the upper branch of the Hadley cell towards the winter hemisphere, but how can subtropical air from the SH penetrate the NH efficiently?_**

A. The difference in transit time from the three source regions to the Arctic is distinct below 450 K, while it is small above 450 K. The efficient transport from the subTR-SH to the Arctic might be driven first by the upper branch of the Hadley cell from the source region to the tropics and then by the quasi-horizontal isentropic transport from the tropics to the Arctic. For the AM air, it is more confined inside the Asian summer monsoon anticyclone in JJA and slowly ascending due to weak diabatic heating rates, and it might take longer time to escape the anticyclone above Asia. We include this change in the manuscript.

(11) **_L210: 'Consequently, air from these three source regions over the Antarctic region exhibits its youngest mean AoA in MAM.' At which levels? (see comment on L185-186)._**

A. Here, it refers to the level below 450 K. It is changed in the manuscript.

(12) **_Figure 5 a-f. There is a marked tilt in the maxima in the NH panels but not in the SH ones. Do you have an idea on what this means?_**

A. The contrast of the marked tilt in the maxima between NH panel and SH panel is not strong for subTR-SH and tropics origin air, and it is distinct for AM origin air. This might be related to the fact that AM origin air is more strongly confined below about 400 K and more rapidly transported to polar regions/high latitudes only above, then the AM origin air sinks downwards slowly at polar regions.

(13) *L231-232: 'The inter-annual variability of AMFs over the southern polar region exhibits a 6-month shift compared to that of the northern polar region.' I cannot see this clearly. Could you point to specific features where this becomes clear?*

A. It refers to the inter-annual variability of AMFs anomaly. For instance, the positive anomalies of AMFs in the sourthern polar region (Fig. 5l) occur six months later compared to those in the northern polar region (Fig. 5i).

(14) *L254: 'underrepresented vertical transport in ClaMS'. How important is this? Why and to what extent is it underrepresented? It would improve the paper if the limitations of the model in this regard were clearly stated. How much does this affect the transit timescales below 2 years?*

A. In our new CLaMS version, which includes an improved representation of convection, particularly for convection unresolved by the ERA5 vertical winds, we observe faster transport from the Planetary Boundary Layer (PBL) to the main convective outflow around $360\,K$ potential temperature. This enhancement significantly improves the representation of CO in CLaMS. In terms of the age spectrum, the contribution of transit times on the order of days to a few weeks is certainly larger than what was discussed in the paper. Initial simulations to quantify the changes in the age spectrum are currently underway. We expect that the small time shifts between $SF_6$ from ACE-FTS and $SF_6$ from CLaMS (shown in Figs. 6a, 6b, 6c, and 6d) can be improved. These shifts are on the order of 1–2 months, with slightly larger differences in the polar SH compared to the polar NH. The new version of CLaMS is becoming operational now, but including this update in the current paper would exceed the scope.

(15) *Figure 7: Why are the relative contributions for subTR-NH and subTR-SH shifted by -10% and +10%, respectively? It does not seem there is a need for this scale shift (I understand the need in the case of the tropical air mass contribution), and it makes the quantitative comparison between panels quite difficult.*

A. This is just for better illustration of the results with the same colorbar. It makes the comparison with AM more direct. The contributions from subTR-SH and subTR-NH are respectively higher and lower than the contributions from AM. The figure for the contribution of subTR-NH almost show orange-red without shift, and the figure for subTR-SH show nearly grey.

(16) *L321-324: Is the difference with previous works relative to inter-hemispheric transport due to more transport in the lower stratosphere (shallow branch of the Brewer-Dobson circulation), or in the upper troposphere Hadley cell? The fact that the tracers are now injected in the boundary layer and not in the upper troposphere may suggest that transport in the troposphere plays an important role, right?*

A. Indeed transport in the troposphere plays an important role here in this study. Substantial Asian monsoon origin air is first transported to the tropics in the troposphere by Hadley cell, and then to the southern hemisphere by the Brewer-Dobson circulation, which makes the contribution to the southern hemisphere larger than that in previous works. Longer simulation period in this study makes the Asian monsoon origin transport to higher levels (above $450\,K$). We revised the manuscript based on reviewer's comment.

(17) *L335-336: 'The easterly phase of the Quasi-Biennial Oscillation (QBO) in 2010 and 2013 (e.g. Anstey et al., 2022) corresponded with positive anomalies in 2010 and 2013 in the northern polar stratosphere and in 2011 and 2014 in the southern polar stratosphere'. Positive anomalies of what?*

A. Positive anomalies of source region air in the polar stratosphere (See Fig.5g-l in the manuscript). We revise the sentence in the manuscript to make the point clear.

(18) **L355-357: 'Additionally, strong downwelling and jet streams in the polar stratosphere play a vital role in isolating stratospheric air masses within the polar vortex during local autumn and winter. Consequently, this phenomenon leads to an increase in AM tracers in the polar stratosphere during these seasons.' This reasoning is confusing: if the polar stratosphere is well isolated why is there 'an increase in AM tracers' ? Rather, in view of the results, I would say that in JJA there is more tropical air in the Arctic stratosphere because of weaker jets and enhanced mixing in the lower stratosphere, while AM air remains mostly confined within the anticyclone. In SON the AM air is liberated from its confinement and can reach polar latitudes. In winter the pole is more isolated so the AM air that was already there remains there. Is that a correct interpretation?**

A. We agree with reviewer's interpretation. Our description seemed to be confusing. Hence, we modified the description accordingly. It is changed as "Additionally, strong downwelling and jet streams in the polar stratosphere play a vital role in isolating older stratospheric air masses within the polar vortex during local autumn and winter. Consequently, this phenomenon leads to an increase in AM tracers in the polar stratosphere below and around the vortex during these seasons." in the manuscript.

(19) **L370: 'Impressively' seems an objective term, change to importantly?**

A. It is changed in the manuscript.

**Referee2**

Many thanks to the reviewer for the comments, they have helped to improve the clarity of the manuscript. In the following, we address all the points raised in the review (denoted by italic letters). Text changes in the manuscript are highlighted in red or blue.

*This paper discusses the transport of surface air from the Asian monsoon region to the stratosphere, particularly the polar regions, based on the ClaMS trajectory model driven by ERA5 data. Overall, the analysis in this paper is of very high scientific quality, and the writing and figures are very clear. The emphasis on dynamics is strong and well-presented. However, in my opinion, the lack of necessary discussion on the chemistry weakens the motivation and overall context of the paper. The lack of adequate motivation necessitates a major revision before acceptance. But I believe this issue is not difficult to address and this manuscript has a potential to be a good one. Please refer to general comment #1.*

**1. General comments**

(1) *In the introduction, lines 49-57, in addition to stating "there is less study on pollutant transport to the polar region" and "CFC-11, HCN, and air can be transported to the polar region," please consider adding a sentence or two to emphasize "how harmful these pollutants are to the polar region and humans." The argument that "this is not studied" is insufficient motivation for a study; "why it is important" should be the key focus.*

*Also, this paper discusses the transport of air, thus using SF6, an extremely long-lived gas. This experimental design is good. However, when discussing "how harmful" these pollutants are, more discussion is needed on pollutants with shorter lifetimes. While a detailed analysis of whether these short-lived pollutants can be transported to the polar region is not necessary for this work, a relevant discussion is important, especially since the first sentence of this work states, "Over the past few decades, rapid economic development in South-East Asia has been associated with a notable increase in the emissions of various pollutants."*

A. Thanks for making us aware of this missing discussion. We have thoroughly worked over the introduction to address these points. For instance, we have added the impacts of these pollutants on the polar region and humans in the same paragraph of the manuscript. "The depletion of the ozone layer caused by CFC-11 and also very short-lived substances (e.g. Brominated substances) leads to increased UV exposure, which can cause skin cancer, cataracts, and immune system suppression in humans(e.g. Bais et al., 2018; World Meteorological Organization, 2022). Ozone depletion is particularly severe in polar regions due to cold temperatures and the presence of polar stratospheric clouds, which accelerate ozone destruction (e.g. Tritscher et al., 2021; World Meteorological Organization, 2022). HCN is highly toxic to humans even at low concentrations. Hence, air pollutants from AM region have potential to affect global environment and human health.".

A short summary is included about the impacts of pollutants on the environment and human health in the second paragraph. "Black carbon and greenhouse gases contribute to global warming, especially in the Arctic, by absorbing sunlight and reducing the reflectivity of snow and ice, leading to accelerated melting and altering climate patterns. Ozone precursors, tropospheric ozone, $SO_2$ and

$NO_x$ cause adverse health effects as well as contribute to the degradation of ecosystems. Exposure to particulate matter and other pollutants can lead to respiratory diseases, cardiovascular problems, and other health issues. The World Health Organization has linked air pollution to millions of premature deaths worldwide. "

(2) *Lines 187-188: "when the tracers are released during boreal summer." I'm confused here.*

*Isn't Figure 3 showing the age of air observed in each season? How can we judge whether the tracers were released during JJA or DJF, etc.? Tracers released in DJF can also be counted when calculating the age of air in JJA, right?*

*Even if you can determine this from Figure 3, is the difference significant? Visually, I can see the difference, but I'm not sure if it is statistically significant. Significance tests are also necessary for other conclusions that compare seasonality.*

A. Yes, the reviewer is right. Figure 3 does show the age of air at the destination region from each season. Tracers released in all the seasons from the source region are counted when calculating the age of air at the destination region. We judge the releasing time of the source tracers based on the transit time. For instance for the age spectrum in DJF, transit times of 1 year, 2 years, 3 years, and so on also correspond to DJF in the source region, whereas transit times of 0.5 year, 1.5 years, 2.5 years, and so on correspond to JJA in the source region.

Figure 3 shows clear peak and nadir of the age spectrum. The large values in "The age spectrum of air originating from the AM region exhibits large values when the tracers are released during boreal summer (JJA)" refer to the peak values in Figure 3. The age spectra of AM origin air in DJF (Figure 3a) show local maxima for transit times around 0.5 years, 1.5 years, 2.5 years, and so on, and they show local minima for transit times around 1 year, 2 years, 3 years, and so on. These peak values correspond to the AM origin air from 0.5 years, 1.5 years, and 2.5 years before DJF, and the nadir values correspond to the AM origin air from 1 year, 2 years, and 3 years before DJF. Hence, the age spectra show large values when the tracers are released during summer. The age spectra show small values when the tracers are released during winter. It is the same for the age spectra during other seasons. For instance, the age spectra of AM origin air in JJA show large values at the transit time around 1 year, 2 years, 3 years, and so on (Figure 3g).

**2. **Specific comments**

(1) *Line 26-27: "the elevated emissions..": please be more specific. In some of your citations, e.g., Rosenlof et al., (1997) and Solomon et al., (2010), they seem to talk about water vapor instead of pollutants. Is water vapor a pollutant?*

A. It is changed to "The elevated emissions of CO, PAN, ozone, HCFC-22, $CH_2Cl_2$, $CH_3Cl$, $CH_3CN$, $CH_3OH$, $HNO_3$, and HCl within the Asian boundary layer have potential impacts on the chemical composition of the atmosphere, thereby influencing atmospheric chemistry, radiative properties, climate, and human health (e.g. Berntsen et al., 1999; Park et al., 2009; Riese et al., 2012; Chirkov et al., 2016; Santee et al., 2017; Rolf et al., 2018; Bian et al., 2020; Adcock et al., 2021; Wang et al., 2022a; Ma et al.,2024)".

(2) *Line 90: "pulsing 40 different species": what are these species? Please list the example of the most important ones for the polar stratospheric chemistry*

A. All the 40 species are artificial pulse tracers in the model for the age spectrum calculation, which represent the air released at different time within 10 years. The first 24 species are pulsed every month for 2 years. The remaining 16 species are pulsed every six month for 8 years. In total, there are 40 species pulsed within 10 years. All the species are passive tracers without chemistry involved in the simulations.

(3) ***Figure 1: why remove 40% mass fraction? Then the conclusion in line 134-135 "2-3 times smaller" is not obvious at all.***

A. We subtract 40% air mass fraction from the tropical origin air to use the same colorbar as AM origin air for direct comparison. Figure RL 1 shows the original air mass fraction from tropics. The plot looks the same as it in the manuscript except the colorbar which is shifted by 40%. The contributions from the source regions to the stratosphere below 450 K is difficult to compare directly. The air mass fraction transport from the AM to the stratosphere above 450 K is around 20%, while it is around 50%-60% for tropical origin air in the stratosphere above 450 K. We revised the text in the manuscript about this.

[Figure]

Figure RL 1: Climatological (2010–2013) zonal mean wind (black contours) and air mass fractions (color shading) originating from the tropics (15° N-15° S) for different seasons.The blue line shows the WMO tropopause.

(4) ***Figure 3& 4: how to explain the stripe pattern of the age of air?***

A. The stripe pattern represents the peak and nadir of air mass fraction transported from the source region to 60° N (Figure 3) and 60° S (Figure 4) at different transit time. For instance, the age spectra of tropical origin air in DJF (Figure 3c) show large values at the transit time around 1 year, 2 years, 3years, and so on, and they show small values at the transit time around 1.5 years, 2.5 years, 3.5 years, and so on. These peak values correspond to the tropical origin air from 1 year, 2 years and 3years

years before DJF, and the nadir values correspond to the AM origin air from 1.5 years, 2.5 years, and 3.5 years before DJF. Please also see the response for the second general comment above.

(5) *Line 201-202: "the pollutants from the source regions released during summer": summer of which hemisphere? The hemisphere of release point, or Antarctica?*

A. It refers to the summer of source region, we revised it in the manuscript.

(6) *Line 230-234 & Figure 5: the interannual variability is very interesting: over polarLS-NH (Figure 5-i), the interannual variability above and below 450 K shows a near opposite pattern, so the interannual variability may from deep branch of the BDC above 450 K and from the shallow branch below 450 K. Over polarLS-SH, the anomaly is consistent throughout all altitudes (Figure 5j-l), and the tilted pattern indicate that the transport of the interannual variability in Figure5j-l is mostly from deep branch of the BDC. Please consider adding corresponding more detailed analysis.*

A. Thanks for pointing to this difference. We agree that these differences are likely due to the combination of BDC and jet differences in both hemispheres. In general, the westerly jets in the southern hemisphere are stronger than those in the northern hemisphere. Hence, the air in the southern polar stratosphere is strongly isolated and more affected by the downwelling of deep branch of BDC. While the air in the northern polar stratosphere is relatively less confined and more strongly affected by the deep branch of BDC, shallow branch of BDC, and quasi-horizontal isentropic transport. We revise this in the manuscript.

(7) *Line 299: "another interesting fact is that the positive SF6 anomaly in the reconstruction over the southern polar region in 2012.." why? is it related to the strength of the polar vortex, BDC, or the AM anticyclone?*

A. We check that the polar vortex in the southern polar region is slightly weaker, and the radiative heating rate indicates stronger downwelling in 2012. The strength of AM anticyclone in 2012 doesn't show extraordinary feature. It also might be affected by ENSO and QBO. We didn't analyze the mechanism thoroughly, it needs more work to confirm the influence of each mechanism. Hence we didn't give a certain explanation about the positive anomaly in 2012.

---

## Author Response (AR2)

Many thanks to the editor for the comments, they have helped to improve the manuscript. In the following, we address all the points raised in the editor report (denoted by italic letters). Text changes in the manuscript are highlighted in red or blue.

*My impression is that you have answer most of the referee comments thoroughly. Therefore, I am pleased to accept your paper for publication subject to minor suggestions as described below.*

(1) *L25: Please define TROPOMI and CAMS*

A. These are changed to "TROPOspheric Monitoring Instrument (TROPOMI) and Copernicus Atmosphere Monitoring Service (CAMS)" in the manuscript.

(2) *L40: SE has not been defined*

A. It is changed to South-East in the manuscript.

(3) *Table 1: The second to last line should say subTR-SH instead of subTR-NH*

A. It is changed in the manuscript.

(4) *L101: Please expand upon the 40 different species as requested / responded to reviewer 2*

A. This part is changed to "The age spectrum $G$ is computed by releasing 240 pulses of inert trace gas species from six distinct source regions, with each region pulsing 40 different species. All the 40 species are artificial pulse tracers in the model for the age spectrum calculation. These pulse tracers approximate a delta distribution lower boundary condition $\chi_0^j$ ($\Omega_i$, t)=$\delta(t - t_j)$, where $j$ ranges from 1 to 40, defining the tracer pulses at specific source times $t_j$. The pulse tracer mixing ratios are initially set to one within the boundary layer of the source region for a duration of 30 days. Outside of the initialization region, these mixing ratios in the boundary are set to zero in each time step. The 40 species are passive tracers without chemistry involved in the simulations, which represent the air released at different time within 10 years." in the manuscript.

(5) *L115. (NOAA)(Dutton et al., 2024) to (NOAA, Dutton et al., 2024)*

A. It is changed in the manuscript.

(6) *L133: The transport contributions from both ... to The transport contributions \*\*to the global stratosphere\*\* from both ...*

A. It is changed in the manuscript.

(7) *L153: as requested by reviewer 1 please explain in the text 'surplus of summertime air' as discussed in the response to reviewers.*

A. This part is changed to "In the tropics, the AM origin air fraction is lower due to wintertime air ascending. However, a lot of AM origin air is transported to the high latitude region during summer driven by the ASM circulation, while the AM origin air is more suppressed and isolated inside the source region during winter. Hence there is evidently a surplus of summertime air compared to wintertime air making its way into the high latitude region." in the manuscript.

(8) *L169 Something is missing in "is primarily limited to the troposphere during boreal, transported ...". Perhaps "is primarily limited to the troposphere during boreal \*\*winter\*\*, transported*

A. It is changed to "is primarily limited to the troposphere during boreal summer, transported ..." in the manuscript.

(9) *L185 Antarctic region nearly exhibits a six-month shift to Antarctic region exhibits a nearly six-month shift*

A. It is changed in the manuscript.

(10) *Figure 5 caption: air mass fractions \*\*(top) or (a-f)\*\* and anomalies \*\*(bottom) or (g-i)\*\**

A. It is changed to air mass fractions \*\*(a-f)\*\* and anomalies \*\*(g-l)\*\* in the manuscript.

(11) *L299: is related to particularly strong transport from AM region (Fig. 7b). \*\*but warrants further investigation\*\*. (or something similar).*

A. It is changed to "is related to particularly strong transport from AM region (Fig. 7b), which warrants further investigation" in the manuscript.

(12) *L361: Note that the QBO was defined in the prior sentence so there is no need to repeat "the Quasi-Biennial Oscillation (QBO" just say QBO*

A. It is changed in the whole manuscript.

(13) *L392-392: Brewer Dobson circulation to BDC*

A. It is changed in the whole manuscript.

(14) *L405 Please add the ACE-FTS data availability.*

A. The ACE-FTS data availability is added in the manuscript.